# Glucocorticoid and dietary effects on mucosal microbiota in canine inflammatory bowel disease

**Todd Atherly[1], Giacomo Rossi[2], Robin White[1], Yeon-Jung Seo[3], Chong Wang[4], Mark Ackermann[5], Mary Breuer[4], Karin Allenspach[1], Jonathan P. Mochel[3], Albert E. Jergens[1]\***

**1** Department of Veterinary Clinical Sciences, College of Veterinary Medicine, Iowa State University, Ames, Iowa, United States of America, **2** School of Biosciences and Veterinary Medicine, University of Camerino, Macerata, Italy, **3** Department of Biomedical Sciences, College of Veterinary Medicine, Iowa State University, Ames, Iowa, United States of America, **4** Department of Veterinary Diagnostic and Production Animal Medicine, College of Veterinary Medicine, Iowa State University, Ames, Iowa, United States of America, **5** Department of Veterinary Pathology, College of Veterinary Medicine, Iowa State University, Ames, Iowa, United States of America

\* ajergens@iastate.edu

**Data Availability Statement:** Raw count data for bacterial counts via FISH and enumerative data for tight junction proteins are found as supplemental files. The data underlying the results presented in

## Abstract

The pathogenesis of canine inflammatory bowel disease (IBD) involves complex interactions between mucosal immunity and the intestinal microbiota. Glucocorticoids are commonly administered to reduce mucosal inflammation and gastrointestinal signs. The study objective was to evaluate the effects of diet and oral prednisone on the spatial distribution of mucosal bacteria in IBD dogs. Eight dogs diagnosed with IBD were treated with immunosuppressive doses of prednisone. The mucosal microbiota from endoscopic biopsies of IBD dogs and healthy controls (HC; n = 15 dogs) was evaluated by fluorescence in situ hybridization (FISH) targeting the 16S rRNA genes of total bacteria and bacterial species relevant in canine/human IBD. Apicaljunction protein (AJP) expression using immunohistochemistry investigated the effect of medical therapy on intestinal barrier integrity. All IBD dogs had a reduction in GI signs following diet and prednisone therapy compared with baseline CIBDAI scores (*P* < 0.05). The mucosal microbiota of HC and diseased dogs was most abundant in free and adherent mucus. Only Lactobacilli were increased (*P* < 0.05) in the adherent mucus of IBD dogs compared to HC. The spatial distribution of mucosal bacteria was significantly different (*P* < 0.05) in IBD dogs following prednisone therapy, with higher numbers of *Bifidobacteria* and *Streptococci* detected across all mucosal compartments and increased numbers of *Bifidobacterium* spp., *Faecalibacterium* spp., and *Streptococcus* spp. present within adherent mucus. Differences in intestinal AJPs were detected with expression of occludin increased (*P* < 0.05) in IBD dogs versus HC. The expressions of occludin and E-cadherin were increased but zonulin decreased (*P* < 0.05 for each) in IBD dogs following prednisone therapy. In conclusion, the spatial distribution of mucosal bacteria differs between IBD and HC dogs, and in response to diet and glucocorticoid administration. Medical therapy was associated with beneficial changes in microbial community structure and enhanced mucosal epithelial AJP expression.

the study are available from DOI: 10.25380/iastate.
10250774

**Funding:** The authors received no specific funding
for this work.

**Competing interests:** The authors have declared
that no competing interests exist.

**Abbreviations:** FISH, Fluorescence in situ
hybridization; UC, Ulcerative colitis; CD, Crohn's
disease; IBD, Idiopathic inflammatory bowel
disease; CIBDAI, Canine inflammatory bowel
disease activity index; GI, Gastrointestinal; GC,
Glucocorticoid; WSAVA, World Small Animal
Veterinary Association; AJP, Apical junction
protein.

## Introduction

Idiopathic inflammatory bowel disease (IBD) is a common chronic enteropathy in dogs char-
acterized by persistent or intermittent gastrointestinal (GI) signs and histopathologic inflam-
mation of the intestines.[1–3] While the exact etiologies for human IBD (i.e., Crohn's disease
[CD] and ulcerative colitis [UC]) remain unknown, current evidence suggests that interactions
between the gut microenvironment (i.e., microbiota, dietary constituents), mucosal immunity
and host genetics initiate and drive chronic intestinal inflammation.[4, 5] Previous studies
have confirmed dysbiosis in the small and large intestines of dogs with IBD that is similar to
altered gut composition observed in human IBD.[6] These shared microbiome changes
include decreases in the phyla *Firmicutes* and *Bacteroidetes* with increases in *Proteobacteria*,
including the Enterobacteriaceae, observed in different studies.[7–9]

Therapeutic strategies for dogs with IBD generally include sequential therapy using spe-
cially formulated antigen-restricted diets, antimicrobials, and immunosuppressive drugs to
induce remission.[3, 10] In instances where affected dogs fail to respond to dietary and antimi-
crobial interventions, immunosuppressive drugs are often administered in the form of gluco-
corticoids (GCs), such as prednisone or prednisolone, being the mainstay of most treatment
regimens.[11, 12] The majority of effects of GCs are attributable to reduced expression of pro-
inflammatory and immune mediators, including cytokines (interleukin [IL]-1, IL-2, IL-6,
tumor necrosis factor-α [TNF-α]) and prostaglandins, which normally amplify intestinal
inflammation.[13] Whether or not GCs beneficially modulate the mucosal microbiota to help
reduce intestinal inflammation in dogs with IBD has not been systematically evaluated using
fluorescence in situ hybridization (FISH).

The objective of this study was to evaluate the effects of hydrolyzed diet and oral prednisone
on the spatial distribution of mucosal bacteria in dogs with IBD. We also characterized the
mucosal expression of tight junction proteins (TJPs) in relation to steroid therapy for induc-
tion of remission.

## Materials and methods

### Ethical animal use

Protocols for endoscopic collection of intestinal biopsies from healthy dogs and dogs with IBD
were approved by the Institutional Animal Care and Use Committee at Iowa State University
(IACUC Log numbers: 1-11-7061; 12-11-7269-K). Written informed consent was obtained
from all owners of enrolled dogs prior to tissue collection.

### Animals

Two groups of animals were studied: dogs with IBD and healthy control (HC) dogs. The IBD
cohort presented with GI signs indicative of enterocolitis. Dogs with IBD (n = 8) were diag-
nosed according to stringent clinical criteria.[2] Enrollment criteria included: (i) a clinical his-
tory of intermittent or chronic GI signs of at least 3 weeks duration; (ii) failed response to
dietary (elimination diet fed exclusively for at least 3 weeks) and antimicrobial (metronidazole
and/or tylosin administered exclusively for 14 days) trials; (iii) failure to document other
causes for gastroenteritis through extensive diagnostic testing; and (iv) histopathologic evi-
dence of mucosal inflammation observed in endoscopic biopsies. Clinical disease activity at
diagnosis and in response to treatment was scored using the CIBDAI.[14] All IBD dogs were
enrolled during a short-term prospective clinical trial of approximately 1 year duration (2015–
2016).

Following diagnosis of IBD, affected dogs were simultaneously placed on a hydrolyzed diet (i.e., Purina HA Hydrolyzed or Royal Canin Canine Hydrolyzed Protein Adult HP to be fed exclusively) and administered oral prednisone at a dosage of 1 mg/kg q12h for 3 weeks then 0.75–0.5- mg/kg q12h for 3 weeks then maintained or tapered over the following 2 weeks (total treatment time = 8 weeks). All dogs with IBD were still receiving both diet and prednisone therapy at the time of repeat GI endoscopy.

The HC group was comprised of 15 young adult dogs (< 3 years of age). None of the dogs had exhibited GI signs for a period of at least 45 days prior to trial enrollment. Moreover, HC dogs were judged to be healthy by one or more physical examinations and on the basis of normal results obtained on CBC and serum biochemical analysis, urinalysis, fecal parasite screens, and dirofilarial antigen testing. All HC dogs were fed a commercial maintenance ration at the time of GI endoscopy. Some of the HC dogs (n = 2) included in the present report were the subject of a previous investigation.[9]

## Endoscopic examination and intestinal biopsy collection

All HC and IBD dogs were subject to routine upper (gastroscopy/duodenoscopy) and lower (ileoscopy/colonoscopy) GI endoscopy for collection of mucosal biopsy specimens. Dogs were prepared for colonoscopy by withholding food overnight and administering an oral polyethylene glycol lavage solution (i.e., GoLYTELY), given twice at a dosage of 20 mL/kg, to evacuate colonic contents. Prior to lower GI endoscopy, the endoscope and pinch biopsy forceps were thoroughly cleaned and sterilized with an activated aldehyde solution and gas sterilization, respectively. Multiple (12–15 mucosal samples from the duodenum; 5–7 mucosal samples from the ileum; 10–12 mucosal samples from the colon) endoscopic biopsy specimens were obtained and fixed in 10% neutral buffered formalin then paraffin embedded for histopathology, using H&E stains, and for FISH. Colonic mucosal biopsies were obtained from each of the ascending, transverse and descending regions, while ileal mucosal biopsies were obtained along the distal 15 cm of this organ. All endoscopic examinations were performed by a single operator (AEJ).

Histopathologic examinations were performed by a single pathologist (MA) blinded as to each dog's health status and clinical course. Mucosal biopsies were assessed for the presence of mucosal inflammation using simplified WSAVA histopathologic guidelines.[15]

## Immunohistochemical evaluation of apical junction proteins (AJPs)

The expression of AJPs was performed to assess intestinal epithelial barrier integrity associated with dysbiosis at diagnosis and in response to diet and steroid administration. In brief, immunohistochemistry for *in situ* expression of AJPs was performed on formalin-fixed colonic biopsy specimens as previously described.[16] Paraffin-embedded tissue sections were rehydrated and neutralized for endogenous peroxidases with 3% hydrogen peroxide for 5 minutes then rinsed for 5 minutes in distilled water. For antigen retrieval, slides were incubated in an antigen retrieval solution of 0.01 M Tris-EDTA buffer (pH9.0) for claudin-2, occludin and E-cadherin in a steamer (Black & Decker, Towson, MD, USA) for 20 minutes. For zonulin stain, slides were immersed in a staining dish containing Sodium Citrate Buffer (10mM Sodium Citrate, 0.05% Tween 20, pH 6.0) which was heated to 95–100˚C in a water bath and with the lid placed loosely on the staining dish for an optimal incubation of 35 minutes. Following incubation, the slides were cooled for 20 minutes then washed in PBS-Tween 20 for 2x2 minutes. For all tissue sections, non-specific binding was blocked by incubation with a protein-blocking agent (Protein-blocking agent, Dako, Carpinteria, CA, USA) for 10 minutes before application of the primary antibodies. Slides were incubated overnight in a moist-chamber (4˚C) with the

following primary antibodies: Polyclonal rabbit anti-claudin-2 (Polyclonal rabbit anti-claudin-2 (PAD: MH44), Invitrogen Ltd., Paisley, UK) and anti-occludin (anti-occludin PAD: Z-T22, Invitrogen Ltd., Paisley, UK) antibodies and monoclonal mouse anti-E-cadherin IgG2α (Monoclonal mouse anti-E-cadherin IgG2α (clone: 36), BD Biosciences, Oxford, UK) as described previously.[16] For zonulin stain, the primary antibody was a rabbit derived poly-clonal antibody (anti-Zonulin pAb, LS-C132998, LSBio Inc., USA, diluted 1:300). The immu-nohistochemistry stain LS-C132998 pAb was validated previously using a panel of 21 formalin-fixed, paraffin-embedded (FFPE) human and canine tissues after heat-induced anti-gen retrieval in pH 6.0 citrate buffer. Following incubation with the primary antibodies, slides were incubated with biotinylated secondary antibodies. These antibodies included: 1) goat anti-rabbit biotinylated immunoglobulin (E0432, Dako, Glostrup, Denmark) used at a dilution of 1:250 and incubated for 1 hour to bind polyclonal rabbit-derived anti-zonulin, claudin-2 and occludin antibodies; and 2) goat polyclonal anti-mouse biotin-coupled secondary anti-body (E 0443, Dako, Glostrup, Denmark) used at dilution of 1:200 and incubated for 1 hour to bind monoclonal murine-derived anti-E-cadherin antibody. The incubation with secondary antibodies was followed by an avidine-biotin complex (ABC *elite*, Vector, Burlingame, UK) incubation of 45 minutes and a chromogen (DAB and Vector VIP, Vector) incubation of approximately 10 minutes, but under direct microscope-control, evaluating the degree of intensity of the stain.

To assess expression of claudin-2, occludin, E-cadherin, and zonulin proteins in endoscopic biopsies obtained before and after diet and steroid treatment, stained tissue sections were eval-uated at ×200 and ×630 (oil immersion) magnification to identify areas of consistent staining and acceptable orientation. Immunostaining was evaluated along the length of multiple enteric/colonic crypts and in areas of intact luminal epithelium. Stain intensity was subjec-tively graded as adequate, and the localization and distribution of chromogen were noted. All 8 dogs with IBD and 10/15 control dogs were investigated for mucosal AJP expression.

All IHC positive cells were quantified using a light microscope (Carl Zeiss), a × 40 objective, a × 10 eyepiece, and a square eyepiece reticule (10 × 10 squares, with a total area of 62,500 μm$^2$). Ten appropriate sites were chosen for quantification and arithmetic means were calcu-lated for each intestinal region. Results were expressed as the number of IHC positive cells per 62,500 μm$^2$. The IHC stained slides were evaluated in a blinded manner by a single pathologist (GR) to confirm staining specificity and to perform quantification of the number of AJP expressing cells.

### Fluorescence in situ hybridization (FISH)

Formalin-fixed ileal and colonic intestinal biopsy specimens (3 μM thick tissue sections) were mounted on glass slides and evaluated by FISH as previously described.[17, 18] In brief, paraf-fin-embedded tissue specimens were deparaffinized using an automated system by passage through xylene (3 x 10 min), 100% alcohol (2 x 5 min), 95% ethanol (5 min), and finally 70% ethanol (5 min). The slides were then transported in deionized water to the DNA testing labo-ratory where they were air dried prior to hybridization. FISH probes 5'-labeled with either Cy-3 or FITC (Life Sciences) were reconstituted with DNAse-free water and diluted to a working concentration of 5 ng/μL (Table 1).

An Eub338 FITC-labeled probe was used for total bacteria counts. For other analyses, spe-cific probes targeting Bifidobacteria, Faecalibacteria, Enterobacteriaceae, Lactobacilli, and Streptococci were labeled with Cy-3 and were applied simultaneously with the universal bacte-rial probe Eub338-FITC. This panel of probes was selected to identify specific bacterial groups and individual bacterial species previously shown to be relevant in the pathogenesis of canine

**Table 1. FISH probe sequences.**

| Probe | Sequence (5' → 3') | Target | Reference |
|---|---|---|---|
| Eub338 | GCT GCC TCC CGT AGG AGT | Total bacteria | Amann (1990) |
| Bif164 | CAT CCG GCA TTA CCA CCC | *Bifidobacterium* spp. | Harmsen (2000) |
| Ebac1790 | CGT GTT TGC ACA GTG CTG | Enterobacteriaceae | Poulsen (1994) |
| Faecali698 | GTG CCC AGT AGG CCG CCT TC | *Faecalibacterium* spp. | Garcia-Mazcorro (2012) |
| Lab158 | GGT ATT AGC ATC TGT TTC CA | *Lactobacillus* spp. | Harmsen (2000) |
| Strc493 | GTT AGC CGT CCC TTT CTG G | *Streptococcus* spp. | Franks (1998) |

IBD.[8, 9] Tissue sections were bathed in 30 μL of DNA–probe mix in a hybridization chamber maintained at 54°C overnight (12 h). Washing was performed using a wash buffer (hybridization buffer without SDS), the slides were rinsed with sterile water, then allowed to air-dry, and mounted with SlowFade Gold mounting media (Life Technologies, Carlsbad, CA) and 25X25-1 cover glass (Fisher Scientific, Pittsburgh, PA).

Probe specificity was confirmed in pilot studies by combining the irrelevant probe non-Eub338-FITC with Eub338-Cy-3, and through hybridization experiments with pure isolates of Bifidobacteria, Fecalibacteria, Enterobacteriaceae, Lactobacilli, and Streptococci to screen for non-selective hybridization.

## Quantification of intestinal mucosal bacteria

Intestinal bacteria were visualized by FISH and 4,6-diamidino-2-phenylindole (DAPI) staining using a 60x Plan Apo oil objective in conjunction with an optional 1.5x multiplier lens on an Eclipse TE2000-E fluorescence microscope (Nikon Instruments Inc., Melville NY) and photographed with a CoolSnap EZ camera (Photometrics, Tuscon, AZ) controlled by MetaMorph software (Nashville, TN). Quantification was only performed when the hybridization signals were strong and could clearly distinguish intact bacteria morphologically by either 2-color (universal and bacterial-specific FISH probe) or 3-color (FISH probes and DAPI stain) identification.

A minimum of 4 different endoscopic biopsy specimens per dog (2 ileal and 2 ascending and/or tranverse colonic biopsies) were evaluated for their mucosal bacterial content. Intestinal tissues (endoscopic biopsies of ileum and colon) for FISH analysis were combined on glass slides for ease of hybridization and consistency of interpretation by a single operator. In this manner, a single slide would contain both ileal and colonic tissues (3–5 samples of each tissue) in which the operator would perform 10 individual bacterial counts in separate endoscopic specimens. While care was taken to accurately distinguish between ileal versus colonic mucosal samples for quantitative analysis, a clear distinction between these tissues could not always be determined. Therefore, mucosal bacterial counts for each animal were recorded as a cumulative total of bacteria observed within ileal/colonic intestinal segments as previously described.[19]

Bacterial quantification was performed in 10 representative fields/dog at a final observed magnification of 600x or 900x. The ten fields included bacteria found within 4 well-defined mucosal compartments: (1) bacteria contained within the mucosa, (2) bacteria attached to the surface epithelium, (3) bacteria localized within adherent mucus, and (4) bacteria found within free mucus.

## Statistical analysis

Sample size of enrolled IBD dogs was dictated by the length of time afforded to the prospective clinical trial. During this period, other dogs (n = 6) were diagnosed with IBD but not enrolled

due to failure to obtain informed client consent or the inability of clients to adhere to specific treatment recommendations.

Mucosal bacteria in the control group (healthy dogs) were compared to the disease group (dogs with IBD) using the Wilcoxon rank-sum (or Mann-Whitney) tests. Differences in bacterial counts between pre- and post-prednisone treatment were assessed using the Wilcoxon signed-rank test. For analysis of the number of cells expressing TJPs, Wilcoxon rank-sum and paired two-sample *t*-tests were performed to investigate statistically significant differences between healthy dogs and dogs with IBD before treatment, and between pre- and post- treatment counts in IBD dogs, respectively. Data analyses were performed using R statistical software (version 3.5.2; The R Foundation for Statistical Computing, Vienna, Austria). *P*-values $< 0.05$ were considered as statistically significant.

## Results

The clinical characteristics of study animals are shown in Table 2. The presenting clinical (gastrointestinal) signs, duration of illness, and severity of endoscopic and histopathologic lesions observed in IBD dogs were similar to previous reports.[1–3, 10, 14, 15] Three dogs with IBD had severe clinical disease activity (i.e., CIBDAI scores $\geq 9$) and 5 dogs with IBD had moderate disease activity (i.e., CIBDAI scores 6–8) at the time of diagnosis (pre-treatment). All IBD dogs had a reduction in GI signs following 8 weeks of diet and prednisone therapy compared with baseline CIBDAI scores ($P < 0.05$). Clinical remission (defined as a $\geq 75\%$ reduction in CIBDAI score from pre-treatment value [11]) was robust by post-treatment at Week 3 and was sustained over the following 5 treatment weeks with tapering dosages of the drug. There was no significant difference in severity of histopathologic scores of mucosal biopsies in IBD dogs pre- versus post-treatment.

The mucosa-associated microbiota of healthy and diseased dogs was most abundant in free and adherent mucus (Tables 3 and 4; S1 and S2 Tables). Sub-populations of bacteria hybridized with probes directed against *Bifidobacterium* spp., *Faecalibacterium* spp., *Lactobacillus* spp., Enterobacteriaceae, and *Streptococcus* spp. There was no significant difference in the number and spatial distribution of the different bacterial species within the mucosa (i.e., attaching to surface epithelia or invasive within tissues) for either HC dogs or dogs with IBD.

**Table 2. Clinical characteristics of study animals.**

| Characteristic | IBD dogs | Controls |
|---|:---:|:---:|
| No. males/no. of females | 5/3 | 0/15 |
| Mean age (yr.) | 5.2 | 2.1 |
| Mean weight (kg.) | 23.1 | 12.1 |
| Mean CIBDAI score[a] | | |
| Pre-treatment score | 7.1 | 0 |
| Post-treatment score | 1.0 | N/A |
| Disease duration (mo.) | 7.7 | 0 |
| Endoscopic lesions[b] | 100% | None |
| Histopathologic lesions[c] | | |
| Mild LPEC | 3 | None |
| Moderate-severe LPEC | 5 | None |

[a] Clinical disease activity, range 0–18.

[b] Mucosal lesions of increased granularity, friability, and/or erosions visualized.

[c] Histopathologic severity of mucosal inflammation; LPEC = lymphocytic-plasmacytic enterocolitis.

**Table 3. Total bacteria via FISH (across all mucosal compartments).**

| Bacterial group | Control dogs Mean ± sd | IBD dogs pre-treatment Mean ± sd | IBD dogs post-treatment Mean ± sd |
|---|---|---|---|
| Bifidobacteria | 28 ± 50 | 38 ± 49 | 240 ± 210[a] |
| Enterobacteriaceae | 42 ± 82 | 148 ± 201 | 124 ± 131 |
| Total bacteria | 329 ± 263 | 500 ± 361 | 950 ± 771 |
| Faecalibacteria | 107 ± 119 | 121 ± 187 | 360 ± 330 |
| Lactobacilli | 16 ± 34 | 31 ± 42 | 41 ± 33 |
| Streptococci | 99 ± 121 | 87 ± 86 | 489 ± 486[a] |

[a] Significant ($P < 0.05$) difference between dogs with IBD pre- versus post-treatment.

Similarly, the total number of Eub338-positive bacteria was not significantly different at diagnosis or following medical treatment in dogs with IBD when compared to HC dogs. Only Lactobacilli were increased (~8 fold, $P < 0.05$) in the adherent mucus of IBD dogs pre-treatment compared to HC dogs. The spatial distribution of mucosal bacteria was significantly different ($P < 0.05$) in IBD dogs following prednisone and dietary therapy, with higher numbers of Bifidobacteria (~6 fold) and Streptococci (~5 fold) detected across all mucosal compartments and increased numbers of *Bifidobacterium* spp. (~30 fold), *Faecalibacterium* spp. (~6 fold), and *Streptococcus* spp. (~20 fold) present within adherent mucus (Figs 1 and 2).

The staining intensity and expression distribution of ileal and colonic AJPs was similar to previous studies.[16, 19] The IHC labeling for all apical junction proteins was most intense along the apical portion of epithelial cells in both ileal and colonic biopsies (Fig 3). Differences in intestinal epithelial barrier proteins were observed with expression of AJP occludin increased (~ 5 fold, $P < 0.05$) in IBD dogs versus HC dogs. The expression of AJPs occludin (~1.5 fold) and E-cadherin (~1.2 fold) was increased but the expression of zonulin decreased (~3 fold, $P < 0.05$ for each) in IBD dogs following prednisone therapy. (Table 5; S3 Table).

## Discussion

The mucosal microbiota in dogs with IBD treated with hydrolyzed diet and oral prednisone for induction of remission was longitudinally investigated using FISH. This study is unique because it compares the mucosal microflora in HC dogs to mucosal bacterial populations in IBD dogs before and after diet and GC administration. We found that dogs with IBD had an

**Table 4. Bacteria present via FISH in adherent mucus compartment.**

| Bacterial group | Control dogs Mean ± sd | IBD dogs pre-treatment Mean ± sd | IBD dogs post-treatment Mean ± sd |
|---|---|---|---|
| Bifidobacteria | 4 ± 9 | 4 ± 6 | 121 ± 147[a] |
| Enterobacteriaceae | 18 ± 59 | 22 ± 32 | 11 ± 6 |
| Total bacteria | 156 ± 139 | 118 ± 133 | 229 ± 174 |
| Faecalibacteria | 33 ± 28 | 23 ± 39 | 135 ± 177[a] |
| Lactobacilli | 0 ± 2 | 8 ± 11[b] | 7 ± 9 |
| Streptococci | 29 ± 35 | 12 ± 18 | 249 ± 398[a] |

[a] Significant ($P < 0.05$) difference between dogs with IBD pre- versus post-treatment.

[b] Significant ($P < 0.05$) difference between control dogs and dogs with IBD pre-treatment.

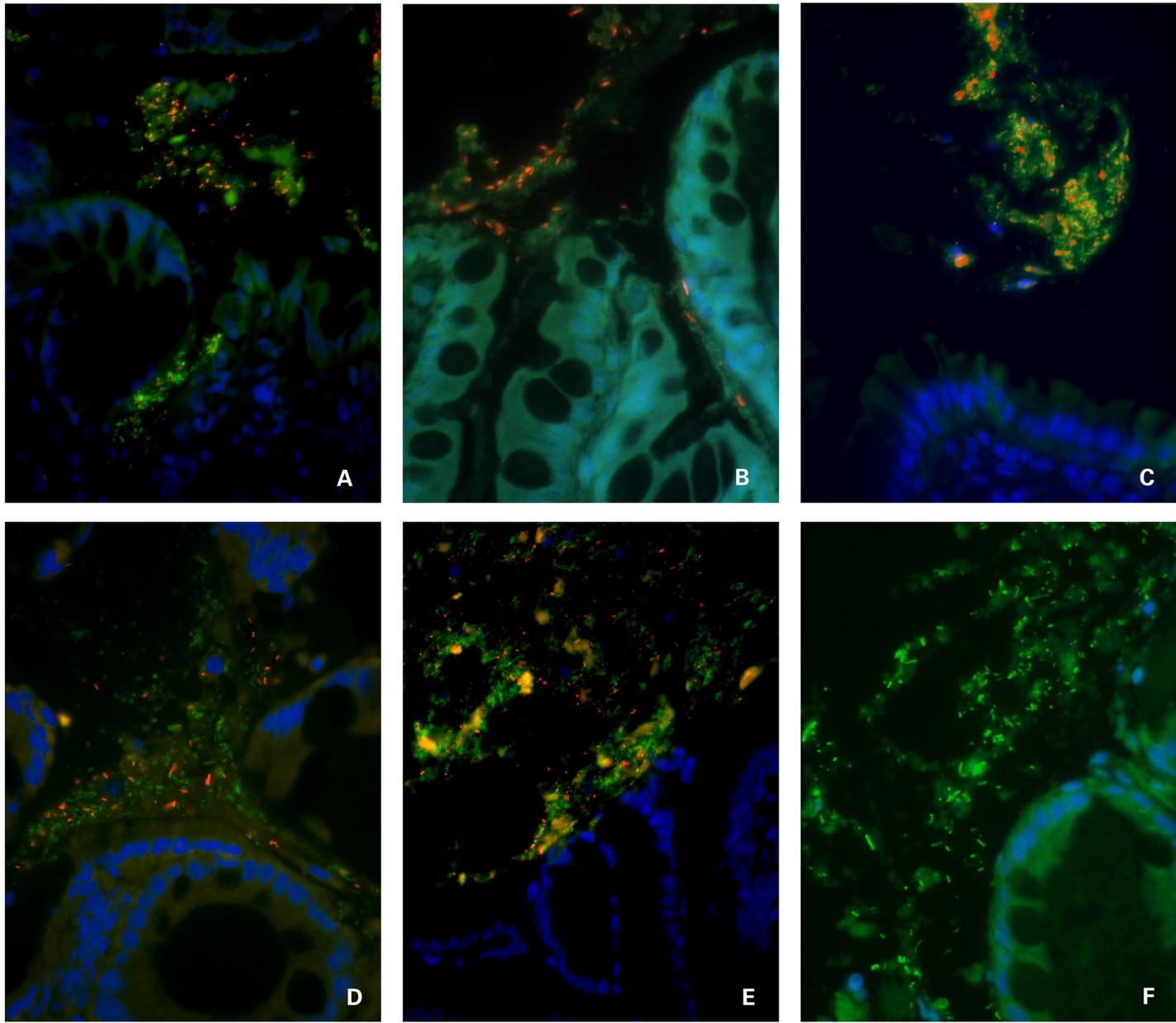

**Fig 1. Triple color FISH identifies mucosal bacteria in the adherent mucus compartment of dogs with IBD following prednisone and dietary therapy.** Panels A-C = ileal tissues and panels D-F = colonic tissues. Panel A = probe Cy3-Faecali698; Panel B = probe Cy3-Ebac; Panel C = probe Cy3- Strc493; Panel D = probe Cy3-Bif164; Panel E = Cy3-Ebac; Panel F = probe FITC-Eub338. All other bacteria that hybridize exclusively with the universal probe (Eub338-FITC) appear green. DAPI-stained intestinal mucosa with goblet cells appears blue. All images at 600x magnification.

increased number of Lactobacilli compared to HC dogs. Changes in the composition and spatial distribution of mucosal bacteria were observed in IBD dogs post-GC and dietary treatment, with more *Bifidobacterium* spp., *Faecalibacterium* spp. and *Streptococcus* spp. found within the adherent mucus compartment. The mucosal expression of different epithelial AJPs varied with increased levels of occludin and E-cadherin but decreased zonulin observed in intestinal biopsies of IBD dogs following prednisone and dietary therapy.

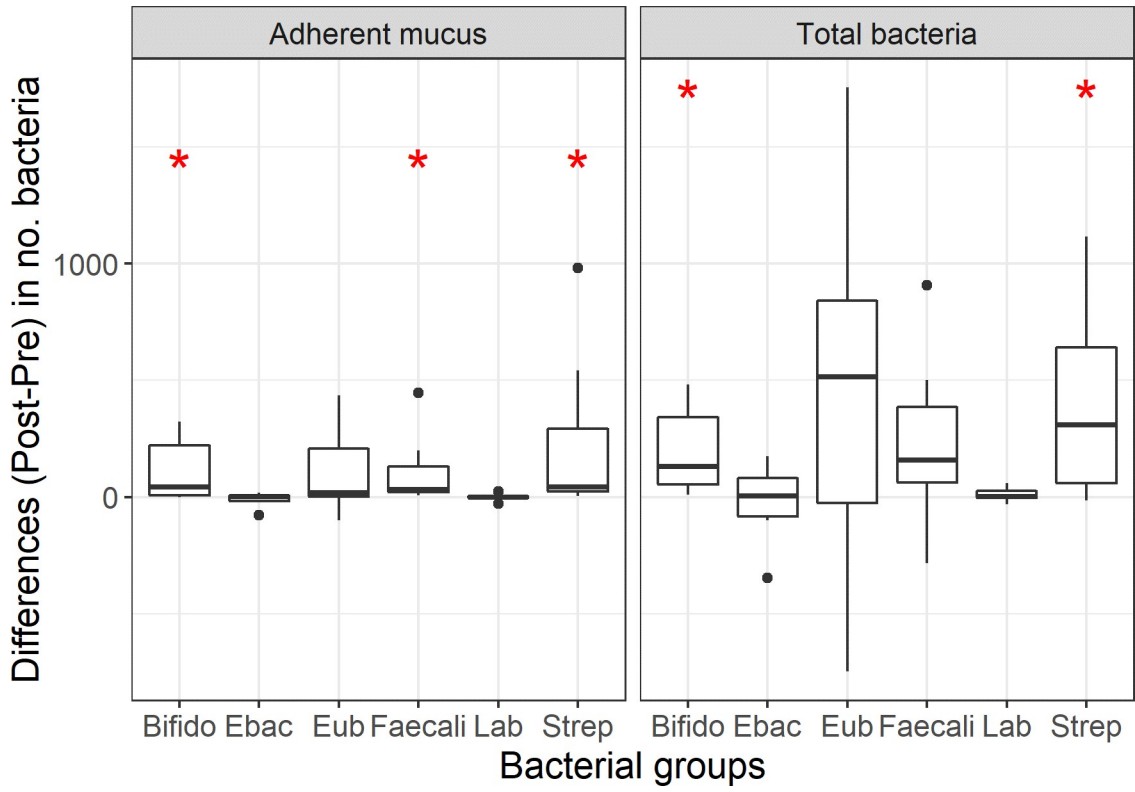

**Fig 2. Box plots showing total mucosal microbiota and microbiota in adherent mucus compartment of dogs with IBD before and after prednisone and dietary therapy.** Differences (*P*<0.05) in the numbers of bacteria between dog groups are indicated by red asterisk.

There is an abundance of clinical evidence implicating alterations of the intestinal micro-biota in the pathogenesis of IBD.[4, 6] Characteristic compositional changes in human IBD patients include decreased microbial diversity with increased numbers of putative harmful bacterial groups (the Enterobacteriaceae, including *Escherichia coli* [*E. coli*], and *Fusobacterium* spp.) combined with decreases in protective species such as *Lachnospiraceae, Bifidobacterium* spp., *Roseburia*, and *Faecalibacterium prausnitzii*.[5, 20, 21] There is very little data describing the microbiome of companion animals with most data derived from the analysis of

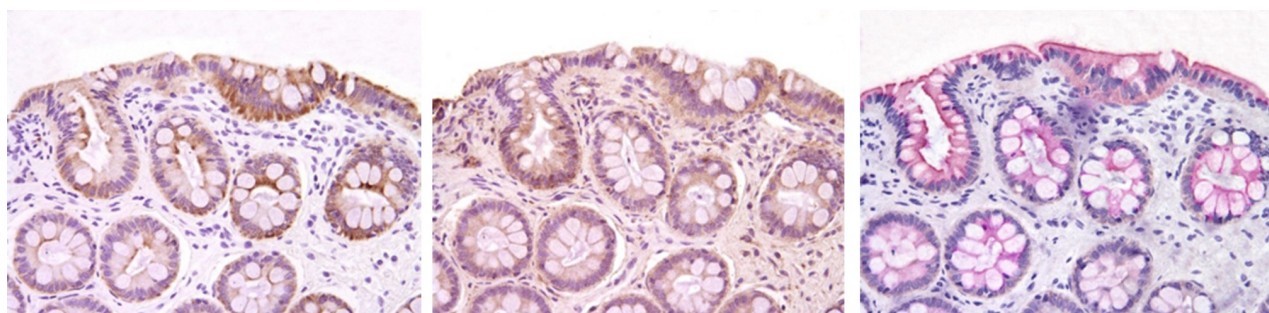

**Fig 3. Immunohistochemical expression of AJPs in colonic biopsies of dogs with IBD after prednisone and dietary therapy.** Protein expression was defined on the basis of cell number and staining intensity of AJPs within the mucosa. Left panel = Claudin-2; center panel = E-cadherin; right panel = occludin. See Table 3 for AJP comparisons between dog cohorts.

**Table 5. Number of intestinal epithelial cells expressing AJPs.**

| Apical junction protein | Control dogs<br>Mean ± sd | IBD dogs pre-treatment<br>Mean ± sd | IBD dogs post-treatment<br>Mean ± sd |
|---|---|---|---|
| Claudin-2 | 1055 ± 75 | 783 ± 493 | 751 ± 454 |
| E-cadherin | 965 ± 230 | 1017 ± 602 | 1287 ± 366[b] |
| Occludin | 181 ± 89 | 1098 ± 324[a] | 1413 ± 399[b] |
| Zonulin | 202 ± 127 | 371 ± 318 | 117 ± 60[b] |

[a] Significant (*P* <0.05) difference between controls and dogs with IBD pre-treatment

[b] Significant (*P* <0.05) difference between dogs with IBD pre- versus post-treatment.

feces. The reported microbial shifts in dogs under disease conditions include lower abundance of *Fusobacteria*, *Bacteroidetes* and *Clostridiales*, but a higher abundance of Proteobacteria in duodenal cytobrushings and duodenal biopsy specimens.[7, 22] In separate studies, FISH techniques have identified invasive *E. coli* in colon biopsies of dogs with granulomatous colitis [23, 24], and increased mucosal-associated Enterobacteriaceae (*E. coli*) in ileum/colon biopsies of dogs with lymphocytic-plasmacytic enterocolitis [9, 19].

Canine chronic enteropathy non-responsive to diet and antimicrobial interventions are designated as idiopathic IBD, which is confirmed by intestinal biopsy results showing mucosal inflammation. In these instances, treatment typically requires immunosuppressive drugs, with systemic GCs often used in most treatment regimens.[12] Separate randomized controlled trials provide strong evidence for administration of oral GCs (i.e., prednisone, prednisolone and budesonide) as induction therapy for dogs with IBD.[11, 25] Still other studies advocate a step-up approach using prednisone first and then adding other immunosuppressive agents, as needed, for non-responsive patients and those having severe clinical disease.[26] A similar strategy is used in human IBD where systemic GCs are administered as first-line therapy for remission induction of mild-moderate Crohn's ileitis/colitis and severe ulcerative colitis.[27, 28]

It is widely accepted that glucocorticoids reduce mucosal inflammation via their generalized anti-inflammatory effects when administered systemically. Broadly, GCs activate many anti-inflammatory genes (i.e., IκB-α [inhibitor of NF-κB], MKP-1 [inhibits MAP kinase pathways], and anti-inflammatory cytokines IL-10, IL-12) and suppress many pro-inflammatory genes (i.e., IL-2, IL-6, IL-13, TNF-α as well as prostaglandins and leukotrienes) that are activated with inflammation.[13] Glucocorticoids also exert post-transcription effects to reduce inflammation by promoting the rapid degradation of pro-inflammatory mRNA (i.e., mRNA derived from TNF-α) and reduced inflammatory protein secretion.[29] Scientific studies describing the effects of glucocorticoids on composition of the intestinal microbiota in healthy animals are scarce. Steroid hormones can modify the composition of the fecal microbiota (primarily the phyla Bacteroidetes and Firmicutes) in rodents following gonadectomy and in response to hormone replacement.[30] Glucocorticoids (corticosterone implant) have also been shown to modulate the fecal microbiome in wild birds by reducing potential pathogenic avian bacteria and members of the Firmicutes.[31] Interestingly, oral administration of prednisolone (1.0 mg/kg daily for 14 days) produced no effect on either bacterial diversity or composition of the fecal microbiota of healthy dogs.[32]

Still other investigations have evaluated the role of glucocorticoids in reshaping the intestinal microbiota of chronic enterocolitis. Using a genetically-susceptible mouse model of colitis, chronic (28 days) administration of dexamethasone was shown to alter fecal microbiota composition (i.e., increased *Bifidobacterium* spp. and *Lactobacillus* spp.) and promote the notable

absence of *Mucispirillum* spp., a gut microbe reliant on mucin.[33] In one small prospective randomized clinical trial, 19 children newly diagnosed with active CD were treated with either enteral nutrition (EN) or corticosteroids (CS) for induction of remission. While both EN and CS induced clinical remission following 8 weeks of continuous therapy, the EN-treated patients showed a change in fecal microbiota composition to a higher proportion of Ruminococcus bacteria (with higher proportions of bacteria belonging to *Clostridium* spp.) as compared to the CS-treated cohort.[34]

A single study has evaluated the intestinal mucosal microbiota in dogs with idiopathic IBD or food-responsive diarrhea (FRD) before and after treatment.[35] Endoscopic biopsies of the duodenum and colon were obtained from 24 dogs (15 FRD, 9 IBD) and evaluated by Illumina sequencing of the bacterial 16S rRNA gene. Dogs with IBD were treated with elimination diet and prednisolone (1 mg/kg BID) for 14 days. Results failed to reveal any significant differences in the overall species richness of dogs with IBD versus FRD. When comparing the effect of treatment on microbiota composition, the duodenum of dogs with FRD was enriched with *Enterococcus* spp., *Corynebacterium* spp. and Proteobacteria pre-treatment, while *Bacteroides* spp. was abundant in the colon of IBD dogs post-treatment.

Our results using FISH confirmed significant changes in the spatial distribution of select mucosal bacteria (i.e., total bacteria and those bacterial sub-populations targeted by the 5 different oligonucleotide probes used in the present study) in dogs with IBD versus HC dogs, and in IBD dogs receiving prednisone and dietary treatment. These data establish that glucocorticoids administered to dogs with IBD alter the distribution and composition of some of the mucosal microbiota (i.e., increased numbers of Bifidobacteria, Faecalibacteria, and Streptococci) in adherent mucus of ileal and colonic tissues which may contribute to induction of remission. The major limitations in the earlier canine study performed by others compared to our report include: i) the absence of a healthy control dog group; ii) failure to describe criteria of clinical remission and treatment group responses relative to changes in microbial composition; iii) failure to include ileal biopsies for microbial compositional analysis, and iv) the use of 16S rRNA gene sequencing which may identify broad shifts in community diversity but provides no information on microbial community structure as does FISH analysis.

It is clear that dysbiosis of the intestinal bacteria is both a cause and consequence of IBD. As antibiotics and probiotics have only modest effects in managing patients with IBD[4], many clinicians now opt for a top-down therapeutic approach using biologics and/or immunosuppressive drugs, such as glucocorticoids, to induce clinical remission[26, 36, 37]. Only few studies show that GCs (or any other immunosuppressive drug for that matter) can exert favorable effects in the fecal microbiome (i.e., increased Clostridia, Bifidobacteria and Lactobacilli) of IBD affected individuals. However, there are no studies (beyond the present report) which describe how or why GCs selectively alter the mucosal microbiota in IBD. It is also possible that other components of the gut microbiome (i.e., fungi and viruses) are influenced by GC administration and contribute to beneficial modulation of resident bacterial populations as observed in this study.[4]

The integrity of the intestinal epithelial barrier is critical for maintenance of mucosal homeostasis. Recent experimental and clinical evidence indicates that apical junction proteins (AJPs), including the tight junction proteins occludin, claudins, and ZO-1 and the adherens junction protein, E-cadherin, provide crucial modulation of epithelial adhesion and intestinal barrier function in IBD.[38] Reports in humans with IBD have demonstrated alterations in AJP expression impacting both paracellular and channel functions of the epithelia.[39] Studies show that claudin-2 abundance often increases in both CD [40] and UC [41] patients while the expression of other barrier-forming claudins may be down-regulated [40, 42] in human IBD. There are conflicts in the expression levels of AJPs in dogs with IBD. Some studies show

increased expression of claudin-2 in colonic tissues at diagnosis [43] or in response to treatment [16], while others report stable baseline occludin, zonulin, and E-cadherin expression [43] or reduced occludin [16] expression in mucosal tissues following combination drug therapy. Our results showed increased expression of AJP occludin in ileal/colonic biopsies of IBD dogs pre-treatment versus HC dogs; with the mucosal expression of occludin and E-cadherin increased but that of zonulin decreased in intestinal biopsies of dogs with IBD post-treatment.

There are some limitations in our study with the first being the small number of IBD dogs enrolled in the trial. A larger cohort of diet and prednisone treated IBD dogs might have yielded more obvious differences in microbial composition as compared to healthy dogs. Our selection of probes used to identify mucosal bacteria was limited and may have missed changes in other microbial community members affected by diet and GC administration. Second, gut microbial populations may potentially vary by age, gender, breed, and dietary consumption. Unfortunately, we were unable to match the age, breed, and sex of HC dogs to dogs with IBD. Also, HC dogs were fed a different (maintenance) ration as compared to the hydrolyzed diet fed to dogs with IBD. It is possible that the varying proportions of dietary constituents (i.e., protein source and content, % fat, etc.) contained in the maintenance and hydrolyzed rations also influenced gut bacterial populations. However, our own studies, evaluating the potential impact of age, body weight, and/or diet, have not identified any associations of microbial abundances with these variables in dogs previously.[9, 44]

In conclusion, the spatial distribution of select mucosal bacteria differs between IBD dogs and HC dogs and in response to diet and glucocorticoid administration. Oral prednisone and dietary therapy were associated with potential beneficial changes in microbial community structure (i.e., increased numbers of *Bifidobacterium* spp., *Faecalibacterium* spp., and *Streptococcus* spp.) and enhanced mucosal epithelial AJP expression, indicating increased epithelial barrier function. Analysis of intestinal microbiota using FISH provides valuable insights on the abundance and spatial distribution of mucosal bacteria which interact most closely with the intestinal epithelium.

## Supporting information

**S1 Table. Denotes actual bacterial counts by mucosal compartment in dogs with IBD pre-versus post-treatment.**
(XLSX)

**S2 Table. Denotes actual bacterial counts by mucosal compartment in healthy (control) dogs.**
(XLSX)

**S3 Table. Denotes actual counts of the number of mucosal cells expressing individual apical/tight junction proteins.**
(XLSX)

## Author Contributions

**Conceptualization:** Albert E. Jergens.

**Data curation:** Todd Atherly, Giacomo Rossi, Robin White, Mark Ackermann, Albert E. Jergens.

**Formal analysis:** Todd Atherly, Giacomo Rossi, Mark Ackermann, Albert E. Jergens.

**Funding acquisition:** Albert E. Jergens.

**Investigation:** Todd Atherly, Giacomo Rossi, Robin White, Albert E. Jergens.

**Methodology:** Todd Atherly, Giacomo Rossi, Albert E. Jergens.

**Project administration:** Albert E. Jergens.

**Supervision:** Albert E. Jergens.

**Validation:** Yeon-Jung Seo, Chong Wang, Jonathan P. Mochel.

**Visualization:** Todd Atherly, Mary Breuer, Jonathan P. Mochel.

**Writing – original draft:** Albert E. Jergens.

**Writing – review & editing:** Giacomo Rossi, Karin Allenspach, Jonathan P. Mochel, Albert E. Jergens.

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
