## [Decision Letter · Decision Letter 0]

10 Oct 2019

PONE-D-19-23630

Glucocorticoid and dietary effects on mucosal microbiota in canine inflammatory bowel disease

PLOS ONE

Dear Dr. Jergens,

Thank you for submitting your manuscript to PLOS ONE. After careful consideration, we feel that it has merit but does not fully meet PLOS ONE’s publication criteria as it currently stands. Therefore, we invite you to submit a revised version of the manuscript that addresses the points raised during the review process.

We would like you to please address the reviewers concerns with special focus on clarifying several concerns as well as improving the presentation of results, figure legends and discussion

We would appreciate receiving your revised manuscript by Nov 24 2019 11:59PM. To enhance the reproducibility of your results, we recommend that if applicable you deposit your laboratory protocols in protocols.io, where a protocol can be assigned its own identifier (DOI) such that it can be cited independently in the future. For instructions see: http://journals.plos.org/plosone/s/submission-guidelines#loc-laboratory-protocols

We look forward to receiving your revised manuscript.

Kind regards,

Pradeep Dudeja

Academic Editor

PLOS ONE

**Journal Requirements:**

**Comments to the Author**

1. Is the manuscript technically sound, and do the data support the conclusions?

Reviewer #1: Partly

Reviewer #2: Yes

2. Has the statistical analysis been performed appropriately and rigorously? 

Reviewer #1: Yes

Reviewer #2: No

3. Have the authors made all data underlying the findings in their manuscript fully available?

Reviewer #1: Yes

Reviewer #2: Yes

4. Is the manuscript presented in an intelligible fashion and written in standard English?

Reviewer #1: Yes

Reviewer #2: No

5. Review Comments to the Author

Reviewer #1: The authors have used two major techniques (IHC and FISH) to determine the effect of glucocorticoid and diet on mucosal microbiota in canine IBD. The manuscript is well written and has very few typographical errors. The authors have provided the raw data that was used for data analysis.

Reviewer #2: Glucocorticoids are commonly administered to reduce mucosal inflammation and gastrointestinal signs.

The study evaluated the effects of diet and oral prednisone on the spatial distribution of mucosal bacteria in IBD dogs. Whether or not GCs beneficially modulate the mucosal microbiota to reduce intestinal inflammation in dogs with IBD has not been systematically evaluated using fluorescence in situ hybridization. The current study should be interesting as it will provide new information in treating IBD dogs and understand the association of inflammation and microbiome. However, the authors need address the following concerns from the reviewer:

1. The sample size is relatively small (IBD dog 8, health control 15). Please clarify the IBD stages of each dog . The focus of the analysis should be the IBD dogs before and after treatment.

2. Please clarify that E-cadherin is not a tight junction protein.

3. The Results are poorly written. More details are needed.

4. The Figure Legends should be separated from the Results.

5. It is helpful to extend the Discussion and relate the current study to human IBD, which is better established.

6. PLOS authors have the option to publish the peer review history of their article (what does this mean?). If published, this will include your full peer review and any attached files.

Reviewer #1: No

Reviewer #2: No

---

## [Author Response · Author response to Decision Letter 0]

15 Nov 2019

RESPONSE TO REVIEWERS (PONE-D-19-23630):

My specific critiques are below: 

• Though the authors use the term ‘spatial distribution of microbiota’ in terms of mucosal compartments, however do not provide any analyses of the difference in distribution between ileum and colon, which were used for FISH analysis. The authors should clarify this and provide data obtained from ileal and colonic samples, since these two intestinal regions have been shown to have different composition of microbiota in various experimental animal species and also in humans

 Intestinal tissues (endoscopic biopsies of ileum and colon) for FISH analysis were combined on glass slides for ease of hybridization and consistency of interpretation by a single operator. In this manner, a single slide would contain both ileal and colonic tissues (3-5 samples of each tissue) in which the operator would perform 10 individual bacterial counts in 4 separate endoscopic biopsies. While care was taken to accurately distinguish between 2 ileal versus 2 colonic mucosal samples for quantitative analysis, a clear distinction between these tissues could not always be determined. Therefore, mucosal bacterial counts for each animal were recorded as a cumulative total of bacteria observed within ileal/colonic intestinal segments rather than separate bacterial counts for the ileum and separate bacterial counts for the colon, respectively. We report a similar study design for FISH analysis in a recent report.1 

 1White, et al. Randomized, controlled trial evaluating the effect of multi-strain probiotic on the mucosal microbiota in canine idiopathic inflammatory bowel disease. Gut microbes. 2017; 8(5):451-66.

 We have now modified the methods section to more accurately reflect how mucosal bacteria in endoscopic samples were counted across intestinal segments.

• Line 381 in discussion: The authors indicate that there’s a change in the distribution and composition of mucosal microbiota, which is misleading. The authors have only looked at five bacteria and their differential expression in the experimental groups. The authors should change the sentence and not extrapolate the observation to differential microbial composition or intestinal site distribution.

 Agreed and now modified in discussion.

• The authors should speculate and discuss why glucocorticoids may alter the differential expression of the three bacteria (line 383)

 Speculation is required here since there is an absolute paucity of information on how GCs may modulate gut mucosal bacterial populations in health or disease. We now state:

 It is clear that dysbiosis of the intestinal bacteria is both a cause and consequence of IBD. As antibiotics and probiotics have only modest effects in managing patients with IBD, many clinicians now opt for a top-down therapeutic approach using biologics and/or immunosuppressive drugs, such as glucocorticoids, to induce clinical remission. Only few studies show that GCs (or any other immunosuppressive drug for that matter) can exert favorable effects in the fecal microbiome of IBD affected individuals. However, there are no studies (beyond the present report) which describe how or why GCs selectively alter the mucosal microbiota in IBD. It is also possible that other components of the gut microbiome (i.e., fungi and viruses) are influenced by GC administration and contribute to beneficial modulation of resident bacterial populations as observed in this study.

• Just for clarity: Were all the biopsies done at the end of 8 weeks of diet and treatment?

 Yes, IBD dogs for repeat GI endoscopy were re-examined during treatment within 1-3 days of diet/drug trial completion.

Review Comments to the Author

Reviewer #1: The authors have used two major techniques (IHC and FISH) to determine the effect of glucocorticoid and diet on mucosal microbiota in canine IBD. The manuscript is well written and has very few typographical errors. The authors have provided the raw data that was used for data analysis.

There are no specific comments to address.

Reviewer #2: Glucocorticoids are commonly administered to reduce mucosal inflammation and gastrointestinal signs.

The study evaluated the effects of diet and oral prednisone on the spatial distribution of mucosal bacteria in IBD dogs. Whether or not GCs beneficially modulate the mucosal microbiota to reduce intestinal inflammation in dogs with IBD has not been systematically evaluated using fluorescence in situ hybridization. The current study should be interesting as it will provide new information in treating IBD dogs and understand the association of inflammation and microbiome. However, the authors need address the following concerns from the reviewer:

1. The sample size is relatively small (IBD dog 8, health control 15). Please clarify the IBD stages of each dog. The focus of the analysis should be the IBD dogs before and after treatment.

Agreed that the focus is on the IBD dogs pre- versus post-treatment. Regarding the various clinical stages of IBD dogs at diagnosis (pre-treatment), 3 dogs with IBD had severe clinical disease activity (i.e., CIBDAI scores > 9) and 5 dogs with IBD had moderate disease activity (i.e., CIBDAI scores 6-8). At trial completion, all 8 dogs with IBD were in full clinical remission with a mean post-treatment CIBDAI score of 1. These new data are now included in the revised manuscript. Good suggestion.

2. Please clarify that E-cadherin is not a tight junction protein.

Thanks for pointing out that there is a difference between the different classes of apical junction proteins (AJPs), which include the tight junction proteins (occludin, claudin family, and ZO-1) and the adherens junction (AJ) protein, E-cadherin. To avoid mis-characterization of the protein classes we evaluated using IHC, we have now corrected the manuscript to state AJPs and their respective proteins rather than TJPs. Good suggestion and sorry for the confusion. We have now updated the discussion section appropriately.

3. The Results are poorly written. More details are needed.

Thank you for this recommendation and we have now expanded the results section regarding patient demographics and FISH analysis to include more detail and clarity as requested. 

4. The Figure Legends should be separated from the Results.

Now separated as requested.

5. It is helpful to extend the Discussion and relate the current study to human IBD, which is better established.

Good suggestion and additional information now provided for the reader. That said, there is no information 

RESPONSE TO REVIEWERS (PONE-D-19-23630):

My specific critiques are below: 

• Though the authors use the term ‘spatial distribution of microbiota’ in terms of mucosal compartments, however do not provide any analyses of the difference in distribution between ileum and colon, which were used for FISH analysis. The authors should clarify this and provide data obtained from ileal and colonic samples, since these two intestinal regions have been shown to have different composition of microbiota in various experimental animal species and also in humans

 Intestinal tissues (endoscopic biopsies of ileum and colon) for FISH analysis were combined on glass slides for ease of hybridization and consistency of interpretation by a single operator. In this manner, a single slide would contain both ileal and colonic tissues (3-5 samples of each tissue) in which the operator would perform 10 individual bacterial counts in 4 separate endoscopic biopsies. While care was taken to accurately distinguish between 2 ileal versus 2 colonic mucosal samples for quantitative analysis, a clear distinction between these tissues could not always be determined. Therefore, mucosal bacterial counts for each animal were recorded as a cumulative total of bacteria observed within ileal/colonic intestinal segments rather than separate bacterial counts for the ileum and separate bacterial counts for the colon, respectively. We report a similar study design for FISH analysis in a recent report.1 

 1White, et al. Randomized, controlled trial evaluating the effect of multi-strain probiotic on the mucosal microbiota in canine idiopathic inflammatory bowel disease. Gut microbes. 2017; 8(5):451-66.

 We have now modified the methods section to more accurately reflect how mucosal bacteria in endoscopic samples were counted across intestinal segments.

• Line 381 in discussion: The authors indicate that there’s a change in the distribution and composition of mucosal microbiota, which is misleading. The authors have only looked at five bacteria and their differential expression in the experimental groups. The authors should change the sentence and not extrapolate the observation to differential microbial composition or intestinal site distribution.

 Agreed and now modified in discussion.

• The authors should speculate and discuss why glucocorticoids may alter the differential expression of the three bacteria (line 383)

 Speculation is required here since there is an absolute paucity of information on how GCs may modulate gut mucosal bacterial populations in health or disease. We now state:

 It is clear that dysbiosis of the intestinal bacteria is both a cause and consequence of IBD. As antibiotics and probiotics have only modest effects in managing patients with IBD, many clinicians now opt for a top-down therapeutic approach using biologics and/or immunosuppressive drugs, such as glucocorticoids, to induce clinical remission. Only few studies show that GCs (or any other immunosuppressive drug for that matter) can exert favorable effects in the fecal microbiome of IBD affected individuals. However, there are no studies (beyond the present report) which describe how or why GCs selectively alter the mucosal microbiota in IBD. It is also possible that other components of the gut microbiome (i.e., fungi and viruses) are influenced by GC administration and contribute to beneficial modulation of resident bacterial populations as observed in this study.

• Just for clarity: Were all the biopsies done at the end of 8 weeks of diet and treatment?

 Yes, IBD dogs for repeat GI endoscopy were re-examined during treatment within 1-3 days of diet/drug trial completion.

Review Comments to the Author

Reviewer #1: The authors have used two major techniques (IHC and FISH) to determine the effect of glucocorticoid and diet on mucosal microbiota in canine IBD. The manuscript is well written and has very few typographical errors. The authors have provided the raw data that was used for data analysis.

There are no specific comments to address.

Reviewer #2: Glucocorticoids are commonly administered to reduce mucosal inflammation and gastrointestinal signs.

The study evaluated the effects of diet and oral prednisone on the spatial distribution of mucosal bacteria in IBD dogs. Whether or not GCs beneficially modulate the mucosal microbiota to reduce intestinal inflammation in dogs with IBD has not been systematically evaluated using fluorescence in situ hybridization. The current study should be interesting as it will provide new information in treating IBD dogs and understand the association of inflammation and microbiome. However, the authors need address the following concerns from the reviewer:

1. The sample size is relatively small (IBD dog 8, health control 15). Please clarify the IBD stages of each dog. The focus of the analysis should be the IBD dogs before and after treatment.

Agreed that the focus is on the IBD dogs pre- versus post-treatment. Regarding the various clinical stages of IBD dogs at diagnosis (pre-treatment), 3 dogs with IBD had severe clinical disease activity (i.e., CIBDAI scores > 9) and 5 dogs with IBD had moderate disease activity (i.e., CIBDAI scores 6-8). At trial completion, all 8 dogs with IBD were in full clinical remission with a mean post-treatment CIBDAI score of 1. These new data are now included in the revised manuscript. Good suggestion.

2. Please clarify that E-cadherin is not a tight junction protein.

Thanks for pointing out that there is a difference between the different classes of apical junction proteins (AJPs), which include the tight junction proteins (occludin, claudin family, and ZO-1) and the adherens junction (AJ) protein, E-cadherin. To avoid mis-characterization of the protein classes we evaluated using IHC, we have now corrected the manuscript to state AJPs and their respective proteins rather than TJPs. Good suggestion and sorry for the confusion. We have now updated the discussion section appropriately.

3. The Results are poorly written. More details are needed.

Thank you for this recommendation and we have now expanded the results section regarding patient demographics and FISH analysis to include more detail and clarity as requested. 

4. The Figure Legends should be separated from the Results.

Now separated as requested.

5. It is helpful to extend the Discussion and relate the current study to human IBD, which is better established.

Good suggestion and additional information now provided for the reader. That said, there is no information

---

## [Decision Letter · Decision Letter 1]

6 Dec 2019

Glucocorticoid and dietary effects on mucosal microbiota in canine inflammatory bowel disease

PONE-D-19-23630R1

Dear Dr. Jergens,

We are pleased to inform you that your manuscript has been judged scientifically suitable for publication and will be formally accepted for publication once it complies with all outstanding technical requirements.

With kind regards,

Pradeep Dudeja

Academic Editor

PLOS ONE

Additional Editor Comments (optional):

Reviewers' comments:

Reviewer's Responses to Questions

**Comments to the Author**

1. If the authors have adequately addressed your comments raised in a previous round of review and you feel that this manuscript is now acceptable for publication, you may indicate that here to bypass the “Comments to the Author” section, enter your conflict of interest statement in the “Confidential to Editor” section, and submit your "Accept" recommendation.

Reviewer #1: All comments have been addressed

Reviewer #2: All comments have been addressed

2. Is the manuscript technically sound, and do the data support the conclusions?

Reviewer #1: Yes

Reviewer #2: Yes

3. Has the statistical analysis been performed appropriately and rigorously? 

Reviewer #1: Yes

Reviewer #2: Yes

4. Have the authors made all data underlying the findings in their manuscript fully available?

Reviewer #1: Yes

Reviewer #2: Yes

5. Is the manuscript presented in an intelligible fashion and written in standard English?

Reviewer #1: Yes

Reviewer #2: Yes

6. Review Comments to the Author

Reviewer #1: (No Response)

Reviewer #2: In the revision, the authors addressed my concerns. There is no further question for this manuscript.

7. PLOS authors have the option to publish the peer review history of their article (what does this mean?). If published, this will include your full peer review and any attached files.

Reviewer #1: No

Reviewer #2: No

---

## [Editor Report · Acceptance letter]

18 Dec 2019

PONE-D-19-23630R1 

Glucocorticoid and dietary effects on mucosal microbiota in canine inflammatory bowel disease 

Dear Dr. Jergens:

I am pleased to inform you that your manuscript has been deemed suitable for publication in PLOS ONE. Congratulations! Your manuscript is now with our production department. 

With kind regards,

on behalf of

Dr. Pradeep Dudeja 

Academic Editor

PLOS ONE